# VITA-E: A Dual-Model Framework for Real-Time, Interruptible, and Concurrent Human-Robot Interaction

## Abstract

Current Vision-Language-Action (VLA) models are often constrained by a rigid, static interaction paradigm, limiting their ability to handle real-time user interruptions or perform concurrent tasks such as speaking while acting. This hinders seamless human-robot collaboration, resulting in an inflexible and unresponsive user experience. To address these limitations, we introduce VITA-E, a novel dual-model framework designed to enable flexible and robust human-robot interaction in real-time. The core of our approach is a dual-model architecture where two parallel VLA instances operate as an "Active Model" and a "Listening Model", allowing one to instantly intervene in the other. We further propose a "model-as-controller" paradigm, where we fine-tune the VLM to generate special tokens that serve as direct system-level commands, coupling the model's reasoning with the system's behavior. Experiments conducted on a physical humanoid robot demonstrate that VITA-E can reliably handle complex interactive scenarios. Our framework is compatible with various dual-system VLA models, achieving a 100% success rate on emergency stops and speech interruptions while also successfully performing concurrent speech and action. This represents a significant step towards more natural and capable robotic assistants.

## 1 Introduction

Thanks to advances in Vision-Language Models (VLMs) (Wang et al., 2024; Zhu et al., 2025; Beyer et al., 2024; Achiam et al., 2023), the field of robotic control has rapidly evolved from task-specific imitation learning (Zhao et al., 2023; Florence et al., 2022; Brohan et al., 2022) towards more general-purpose multi-task action generation (Zitkovich et al., 2023; Kim et al., 2025; Ghosh et al., 2024) and open-ended instruction following (Shi et al., 2025; Bjorck et al., 2025; Huang et al., 2025; Zawalski et al., 2025), bringing us closer to the goal of a general-purpose robot. However, the predominant focus of the field has been on improving the success rate of specific, static tasks, often overlooking a critical dimension of autonomy: the ability to engage in continuous, natural, and dynamic collaboration with a human user in complex scenarios (Abbo et al., 2025; Fong et al., 2003). An ideal robotic assistant should not be a silent executor of commands but a collaborative partner, capable of communicating its state while acting (e.g., answering, "Is the bookshelf tidied up?" while organizing a room) and dynamically adapting to new directives that reflect a changing environment (e.g., "Don't clean the bedroom yet—the baby is sleeping.").

Although some work has begun to address complex instructions and human feedback (Shi et al., 2025; Ahn et al., 2022; Shi et al., 2024; Li et al., 2025), existing systems are fundamentally constrained by a rigid and static interaction paradigm. This paradigm imposes three critical limitations that prevent flexible human-robot collaboration: 1) **Uninterruptibility:** The robot becomes locked into its current action or a lengthy spoken response, unable to be interrupted by the user's immediate needs, which forces unnatural pauses in the interaction. 2) **Lack of Concurrency:** Systems typically cannot perform physical actions and engage in verbal dialogue simultaneously, limiting their efficiency and ability to multitask in a human-like manner. 3) **Interaction inflexibility:** Together, these constraints create a stilted, unresponsive experience where the inability to interrupt or act concurrently makes the interaction feel slow and awkward, resulting in a high perceived latency.

To break through this paradigm bottleneck, we design and implement VITA-E, a system architected specifically for flexible, real-time human-robot interaction. As shown in Figure 1, VITA-E is capa-

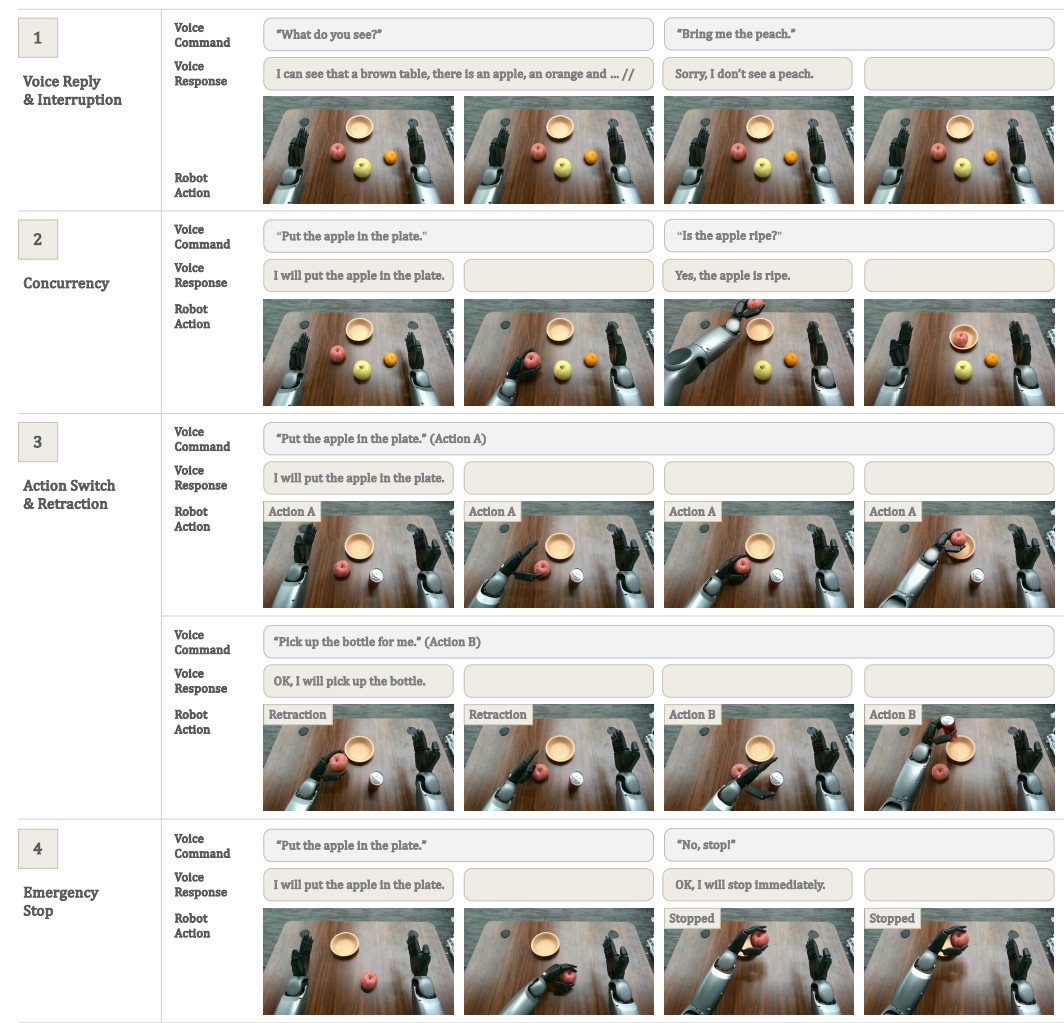

Figure 1: VITA-E is capable of handling a variety of complex interactive scenarios, including real-time interruption and concurrency.

ble of handling these complex interactive scenarios. Inspired by the cooperative mechanism of the human brain's hemispheres and the interruptible model VITA (Fu et al., 2024), our framework features a novel dual-model interaction core. In this design, one model instance acts as the "executing hemisphere," focused on the current task, while the other acts as a "listening hemisphere", ready to process new user requests. This parallel architecture fundamentally overcomes the limitations of sequential execution, providing the foundation for true real-time interaction.

Our primary contributions are as follows:

- **A Novel Dual-Model Interaction Core:** We propose a parallel processing architecture where two VLA instances work in concert. One acts as an active model for the current task, while the other serves as a listening model, enabling simultaneous speech and action, as well as the instant interruption of any ongoing task.

- **A Special Token-Based Control Flow:** We design a set of special tokens (e.g., `[ACT]`, `[HALT]`) that are generated by the VLM itself to directly drive the system's state transitions. This "model-as-controller" paradigm creates an elegantly tight coupling between action and system behavior.

- **A Methodology for Training Interactive VLAs:** We introduce a data curation and fine-tuning strategy to teach a VLM to generate system-level control tokens. Our methodology is implemented upon a mainstream VLM + Diffusion Action Expert architecture, demonstrating its compatibility with a wide range of state-of-the-art dual-system models.

## 2 RELATED WORK

### 2.1 FOUNDATION VLA MODELS

The powerful understanding and generalization capabilities of VLMs have significantly boosted the development of VLA models. A prevalent and effective approach (Zitkovich et al., 2023; Kim et al., 2025; Ghosh et al., 2024; Wu et al., 2024) involves directly fine-tuning a pre-trained VLM on specific robotics trajectory datasets, teaching it to output corresponding action tokens based on environmental context and language instructions. RT-2 (Zitkovich et al., 2023) tokenizes actions into the same format as text and co-trains the model on a mixture of vision-language and robotic control datasets, enabling it to directly generate action tokens for motor commands. Following this paradigm, OpenVLA (Kim et al., 2025) further explores efficient fine-tuning strategies for VLA models. Although these end-to-end methods can fulfill simple commands, this approach risks degrading the VLM's native vision-language understanding and reasoning capabilities, often leading to sub-optimal performance on complex tasks.

To mitigate this issue, a decoupled dual-system architecture (Black et al., 2024; Bjorck et al., 2025; Yuan et al., 2025; Huang et al., 2025; Chen et al., 2025; Lin et al., 2025) has been widely explored, where "System-2" performs high-level tasks and scene understanding and "System-1" translates relevant information into low-level, executable actions. $\pi_0$ (Black et al., 2024) appends a diffusion-based action expert to a pre-trained VLM, allowing the model to inherit robust vision-language understanding from web-scale data while achieving precise manipulation control. Similarly, GR00T (Bjorck et al., 2025) adopts a comparable architecture and augments its training data by learning latent action representations from demonstration videos, which enhances its performance and generalization. However, these state-of-the-art models typically assume that user instructions are provided once at the beginning of a task and remain static, thereby overlooking the dynamics of human-robot interaction.

### 2.2 INTERACTIVE VLA SYSTEMS

Considering the fluid nature of human intent and the inherent limitations of current models, the ability to perceive and adapt to new instructions timely is crucial. SayCan (Ahn et al., 2022) combines the high-level semantic understanding of large language models with learned robotic affordances, allowing robots to execute complex, multi-step natural language instructions. VILA (Hu et al., 2023) uses vision-language reasoning for long-horizon task planning and integrates natural visual feedback to perform complex long-term tasks. RT-H (Belkhale et al., 2024) and YAY Robot (Shi et al., 2024) introduce an intermediate-language-based action hierarchy, where high-level policies generate instructions that guide low-level policy execution, permitting human intervention and adjustment via natural language. RACER (Dai et al., 2024) achieves task execution and error correction through a supervised-executor dual-model architecture. Recently, Hi-Robot (Shi et al., 2025) makes progress in this direction by using a high-level VLM to translate multi-stage, complex instructions into simplified atomic steps for a low-level VLA model to execute. During execution, the high-level VLM can process new instructions and adjust the control policy for the next step after the current atomic step has been completed. Another work, Switch-VLA (Li et al., 2025), incorporates contact state and behavior modes to seamlessly achieve task switching based on the latest instruction.

Nevertheless, the interactivity of these approaches is still constrained. They often cannot be interrupted mid-action and must complete their current atomic action or inference cycle before processing a new user directive. This introduces significant latency and limits the system's real-time responsiveness and flexibility. Although Switch-VLA achieves faster response by considering language commands at every action generation step, this design constrains the size of the VLM that can be used, ultimately limiting its capabilities and performance. In this paper, inspired by and based on the interruptible full-duplex voice interaction system VITA (Fu et al., 2024; 2025), we propose VITA-E, a novel VLA framework that supports large-scale models while enabling fluid real-time interaction, concurrent speech and action, and the instant interruption of any ongoing task.

## 3 VITA-E SYSTEM ARCHITECTURE

This section presents the core technical contribution of our work, detailing the architectural and algorithmic innovations that enable flexible human-robot interaction.

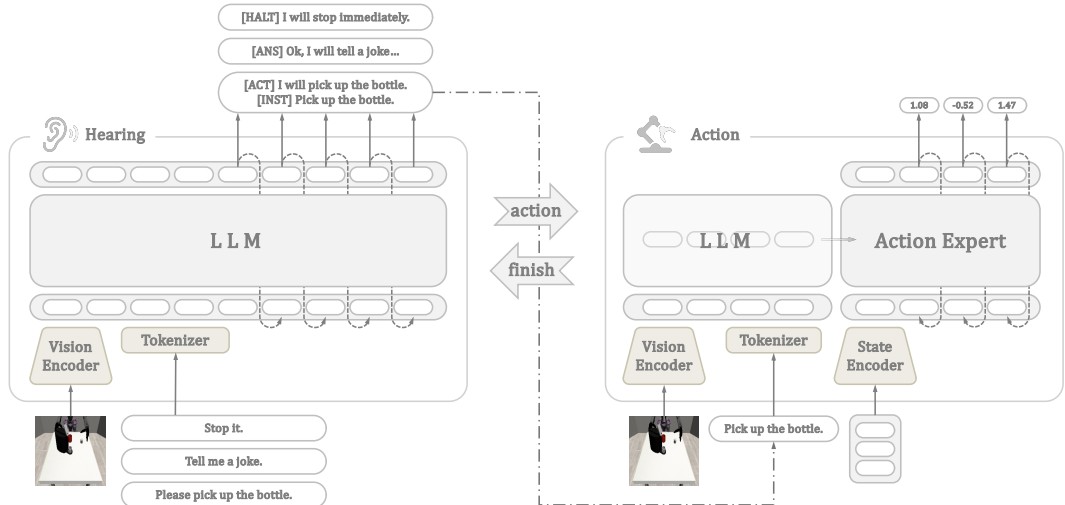

Figure 2: The logical architecture and operational states of the active model in our dual-model VITA-E framework. Each of the two models can switch between "Active" and "Listening" states. When a model becomes active, the VLM part acts as a controller, processing user inputs in the "Hearing" state and generating special tokens that can trigger a transition to the "Action" state, where it collaborates with an action expert as a whole VLA model. For example, when the VLM's output starts with the [ACT] token, the text between the [ACT] token and the [INST] token will be played as audio, and the text after the [INST] token will be sent to the VLA model as an action instruction. Otherwise, the text after the special token will only be played as audio, and the system will execute the command corresponding to the special token. For the detailed functions of each special token, see Table 1.

### 3.1 OVERALL FRAMEWORK

The VITA-E framework is designed around two foundational principles: 1) a **model-as-controller** paradigm, where a VLM drives system behavior by generating explicit command tokens, and 2) a **dual-model interaction core** that enables real-time interruption and concurrency.

Our framework adopts a dual-system architecture that has recently become common (Bjorck et al., 2025; Black et al., 2024)), which consists of a VLM for high-level understanding (System-2) and a diffusion action expert for low-level motor control (System-1). The VLM's primary role is to interpret the user's intent and the scene context, while the action expert translates this understanding into precise physical movements. The logical flow of our approach is illustrated in Figure 2. It operates primarily in two states: **Hearing** and **Action**. In the default **Hearing** state, the VLM processes image and user language to determine intent. Based on its understanding, it can generate a [RES] token for a purely verbal response, or an [ACT] token, if a physical task is commanded. The generation of this [ACT] token serves as a direct command, transitioning the system into the **Action** state. In this state, the VLM works with an Action Expert to generate low-level motor commands until the task is finished (signaled by an [END] token) or interrupted by the other model.

This entire logical framework is implemented through a modular server-client implementation, where the server hosts our dual-model core and the client captures the real-world information and executes the action command. The following subsections provide details of: 1) the model-as-controller paradigm enabled by the VLM and its special token-based control language, and 2) the dynamic interaction mechanism enabled by the dual-model core.

### 3.2 THE MODEL-AS-CONTROLLER PARADIGM

#### 3.2.1 PROBLEM FORMULATION

The goal of our framework is to learn a policy that maps the current sensory inputs and user instructions to a sequence of robot actions, while also generating real-time verbal responses and handling interruptions. Formally, at each timestep $t$, the system receives a visual input $I_t$, the robot's propri-

oceptive state $q_t$, and a natural language instruction from the user $L_t^{\text{user}}$. The objective is to produce a verbal response $L_t^{\text{robot}}$ for the user, a system behavior control $c_t$, and a corresponding action chunk $A_t$ for the robot to execute.

### 3.2.2 THE CONTROL LANGUAGE: VLM WITH SPECIAL TOKENS

Our key innovation is to have the VLM produce not only semantic understanding but also explicit system-level commands via a learned "control language." The VLM, denoted as $\pi_{\text{VLM}}$, consumes the visual input $I_t$ and user instruction $L_t^{\text{user}}$ and outputs a structured string $S_t = (c_t, L_t^{\text{robot}}, C_t^{\text{robot}})$. Here, $c_t$ is a discrete control token (see Table 1), $L_t^{\text{robot}}$ is the verbal response, and $C_t^{\text{robot}}$ is an action command that is absent ($\varnothing$) unless $c_t$ indicates that robot action is required. The VLM is trained to learn $\pi_{\text{VLM}}(S_t \mid I_t, L_t^{\text{user}})$.

Table 1: Special tokens used to control the VITA-E framework's behavior.

| Token | Description | Example Model Output |
|-------|-------------|----------------------|
| [RES] | Signals a voice-only response. Generated as the first token for conversational replies. | [RES] I see an apple on the table. |
| [ACT] | Signals that the response includes a physical action. Generated as the first token to enter action mode. | [ACT] Okay, I will put the toy in the box. [INST] Pick up toy and place in box. |
| [INST] | Delimits the spoken part of an action response from the internal action instruction that follows. | |
| [HALT] | Commands an immediate stop of the current action. Generated as the first token for emergency stops. | [HALT] Stopping immediately. |
| [END] | Signals that a multi-step action sequence has been successfully completed. | [END] The action is finished. |

Teaching the VLM to generate these control tokens required a specialized data curation process that goes beyond standard instruction-following datasets. We reformat and synthetically augment the embodied scenario vision-language data to explicitly teach the VLM to output control tokens as desired, based on datasets including ActionNet (Team & Mu, 2025), Libero (Liu et al., 2023), and self-collected real-world scenario data.

The process is as follows: we first process our aggregated dataset of demonstration trajectories, which initially consists of video, instructions, and actions. We then apply an automated annotation pipeline to insert the special tokens into the target output based on the context of each trajectory:

- For trajectories where the instruction is a question (e.g., "What do you see?") and no significant action occurs, the target response is prepended with the [RES] token.

- For trajectories involving physical manipulation, the target response is prepended with [ACT]. A generic spoken confirmation (e.g., "Okay, I'll do that.") is generated, followed by the [INST] token and a cleaned version of the original instruction.

- To create training instances for interruption, we take an existing action trajectory and inject a new user input like "Stop!" at a random point. The corresponding target output for this synthetic data point is then formulated as "[HALT] Okay, stopping.".

- Finally, to teach the model to signal task completion, we use the final state of successful trajectories. At the point where the task is complete and after, we create training instances where the target output begins with the [END] token.

This data curation strategy transforms the fine-tuning task. Instead of merely learning to describe or plan actions, the VLM learns to output a structured string that simultaneously contains a conversational reply, a system-level command (the special token), and a semantic goal for the action expert. This approach is what enables the tight coupling between high-level reasoning and low-level system execution that defines our framework.

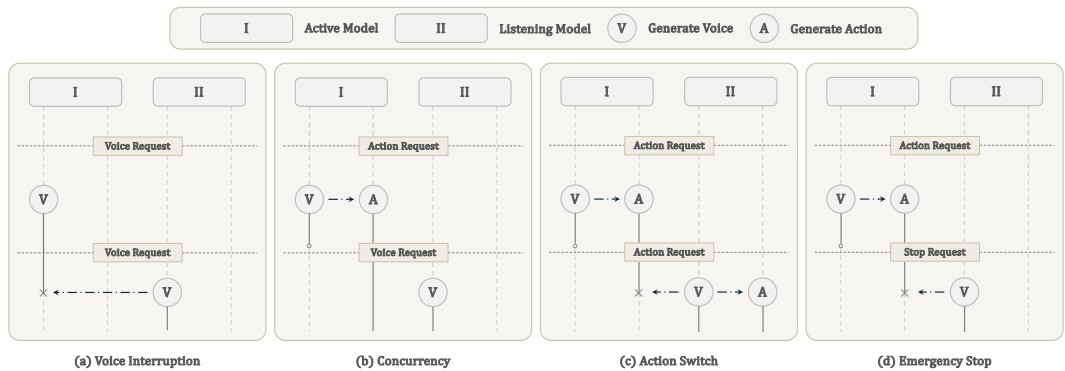

Figure 3: Sequence diagrams illustrating the four primary interaction modes. Model I is the Active Model, and Model II is the Listening Model. V and A represent voice and action generation, respectively. The Listening Model can process new requests in parallel or preempt the Active Model to handle interruptions and task switches.

### 3.2.3 ACTION EXPERT FOR MOTOR CONTROL

The action expert can be denoted as $\pi_a$, and the generation process of the action chunk will be $A_t = \pi_a(h_t, q_t)$, where $h_t = \pi_{\text{VLM}}(I_t, C_t^{\text{robot}})_{\text{hidden}}$ is the hidden states of the VLM. We adopt the Diffusion Transformer from GR00T (Bjorck et al., 2025) as $\pi_a$, which has been pre-trained on large-scale embodied data, providing a strong foundation for motor control. Following our two-stage training paradigm, we further fine-tune this model on task-specific data collected from our target robot. During this stage, we exclusively train the projection head. This approach adapts the model to the specific kinematics of our robot while preventing overfitting. The action expert is trained to predict a sequence of robot joint angles, translating the VLM's high-level semantic goal into low-level, executable trajectories.

### 3.3 DYNAMIC INTERACTION VIA A DUAL-MODEL CORE

The core of VITA-E's interactivity lies in its unique dual-model architecture, a design inspired by the cooperative mechanism of the brain's hemispheres. This "two hemispheres" approach allows one model instance—the **Active Model**—to remain focused on executing the current task, while the other instance—the **Listening Model**—serves as an observer. This structure enables the robot to instantly shift its attention to achieve flexible concurrency and seamless interruptions. The coordination between these two hemispheres is managed by synchronization primitives (e.g., semaphores) that control which model is active and, crucially, grant the Listening Model the authority to intervene in its counterpart. The four primary interaction patterns that emerge from this architecture are illustrated in Figure 3.

**Voice Interruption** As shown in Figure 3(a), the framework allows users to interrupt the robot's speech at any time. While the Active Model (I) is generating a verbal response to a `Voice Request`, any subsequent input will be routed to the Listening Model (II). The Listening Model's priority is to process this new input: it signals a preemption event that instantly terminates the Active Model's speech generation, ensuring the robot yields to the user for flexible, natural turn-taking.

**Concurrency** VITA-E is able to perform physical actions and speak simultaneously, as depicted in Figure 3(b). When the Active Model (I) is engaged in an `Action Request`, it enters a protected state. If a new `Voice Request` arrives, the Listening Model (II) assesses the Active Model's state. Recognizing that an action is in progress, it proceeds to handle the conversational query independently by generating its own voice response, all without interrupting the ongoing physical task. This allows the robot to answer questions while continuing its work.

**Action Switching** The framework can dynamically switch from one physical task to another based on user commands, as shown in Figure 3(c). If the Active Model (I) is executing an initial `Action Request` (Task A), the subsequent new instruction for Task B will be captured by the Listening Model (II). The Listening Model then preempts the Active Model, halting its execution of Task A. Immediately after, the Listening Model becomes the new Active Model and executes Task B. This allows for a seamless switch of the robot's physical behavior. To ensure safety during this transition,

a retraction mechanism is employed, which returns the robot to the initial pose by sequentially popping and executing inverse movements from a stored action stack.

**Emergency Stop** Another interactive feature is the ability to immediately halt all motion. As illustrated in Figure 3(d), when a user issues a `Stop Request` while the Active Model (I) is performing an action, the Listening Model (II) processes it with the highest priority. It generates a `[HALT]` token and triggers an immediate preemption of the Active Model, which breaks its action-generation loop and sends a final halt command to the robot client. This ensures that all physical motion ceases instantly, providing a reliable and highly responsive safety mechanism.

## 4 EXPERIMENTS AND RESULTS

In our experiments, we evaluate the VITA-E framework's capabilities across a set of interactive scenarios. We specifically aim to demonstrate its proficiency in: 1) executing fundamental manipulation tasks, 2) providing timely spoken responses, 3) performing actions and speech concurrently, 4) handling dynamic task switching, and 5) responding to emergency stop commands.

Our experiments are conducted on a Fourier GR2 humanoid robot platform (Fourier Team). The system's input consists of visual data from a first-person-view static Realsense D455 RGB camera mounted on the robot's head, along with the robot's proprioceptive states.

### 4.1 FUNDAMENTAL MANIPULATION TASKS

**Simulation experiments.** To establish a baseline for manipulation capability, we first evaluate the model's performance on the Libero benchmark. The Libero benchmark is designed to evaluate a model's ability to apply and transfer action skills, which is composed of four action task suites: Spatial, Object, Goal, and LONG.

We compare our model on the Libero benchmark with the baseline model GR00T (Bjorck et al., 2025). For a fair comparison, we only replace the VLM of the Eagle-2 model in GR00T with the VITA-1.5, and freeze the VLM part, fine-tuning only the diffusion action expert. We first pre-train the model on the Libero-90 dataset, and then finetune it on the mixture of the four action task suites. We report the average success rate of these two models in Figure 4.

Experimental results show that our model successfully completes a majority of tasks in the Libero benchmark. However, we acknowledge a performance gap when compared to the GR00T model. While VITA-E excels at object recognition, it exhibits limitations in its understanding of spatial relationships and goal concepts. It is crucial to note that GR00T unfreezes the visual encoder and aligner parameters of its Eagle-2 VLM and jointly optimizes them with the diffusion action model via end-to-end training on a large-scale embodied dataset. In contrast, VITA-E does not leverage such large-scale pre-training and is trained with a totally frozen VLM. Therefore, the observed disparity in success rates is an expected outcome given these significant architectural and data-related constraints.

**Real robot experiments.** To demonstrate the model's capabilities on the real robot, we evaluate the model's fundamental pick-and-place skills across two scenarios: 1) picking up a can from the table, and 2) picking up a toy and placing it into a basket. For each task, we collect 300 demonstrations by teleoperating the robot arm at 20 Hz across 26 degrees of freedom.

To show the performance of our method, we select several state-of-the-art models and train them on our collected dataset. For $\pi_0$ (Black et al., 2024) and Diffusion Policy (Chi et al., 2023), we train the entire model. For GR00T (Bjorck et al., 2025) and SmolVLA (Shukor et al., 2025), we train their entire action expert modules. In contrast, and to mitigate overfitting, we fine-tune only the action projector of VITA-E. We evaluate each method for 30 trials to obtain the final success rate, as reported in Figure 5.

The results demonstrate that VITA-E is a highly capable manipulation model, whose performance is on par with state-of-the-art models. While the goal is not to outperform the top manipulation-focused baselines, these results confirm that our system's core capabilities are sufficiently strong to reliably evaluate our primary contribution: a novel architecture for real-time human-robot interaction. Moreover, it is equally worth emphasizing that our architecture remains compatible with dual-system VLA models such as $\pi_0$.

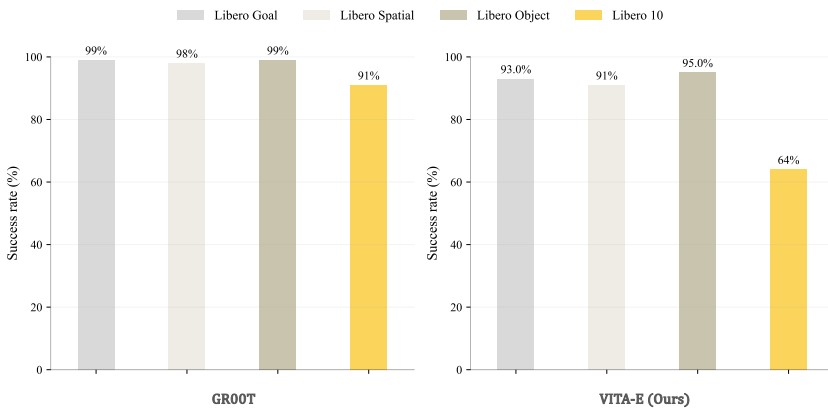

Figure 4: Success rate comparison of VITA-E and GR00T on the Libero benchmark. Although the success rates of VITA-E on this benchmark are not as good as the baseline with the same structure, we want to emphasize that the goal of this work is not for VITA-E to achieve the state-of-the-art experimental performance, but to prove that the model is capable of completing embodied tasks and to provide quantitative metrics.

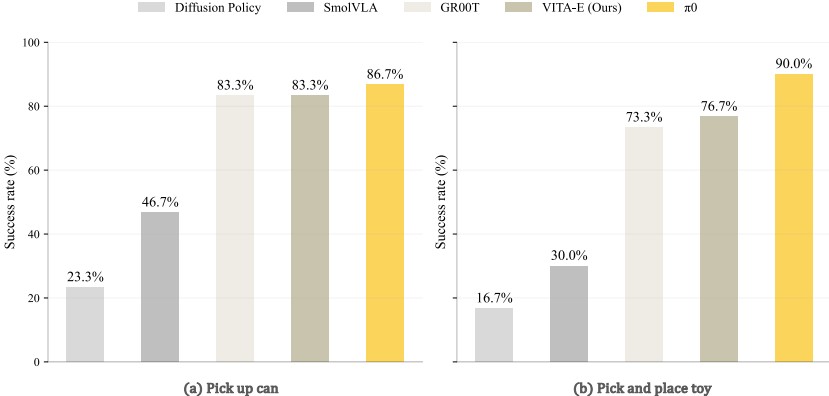

Figure 5: Success rate comparison of VITA-E and baseline models on two fundamental manipulation tasks: (a) Pick up can and (b) Pick and place toy. Results are reported over 30 evaluation trials.

## 4.2 INTERACTIVE TASKS

The core contribution of our work lies in enabling flexible and robust human-robot interaction. We evaluate VITA-E on four critical interactive capabilities: speech-action concurrency, speech inter-ruption, action switching, and emergency stops. We evaluate the concurrency skill qualitatively, while we measure the other three by their success rate in 30 trials. As the baseline methods evalu-ated in the previous section are not designed with analogous real-time interaction capabilities, our evaluation of these tasks focuses exclusively on our proposed framework.

For the concurrency task, we observe that VITA-E can consistently provide spoken answers to user queries while smoothly continuing its manipulation task, without any noticeable pauses or degra-dation in its physical execution. The average latency of VITA-E's voice responses across ten tests is 2.26s. The quantitative results for the other tasks are reported in Table 2. The results show that VITA-E achieves a perfect 100% success rate in both interrupting its own speech and in the emergency stop task. This near-perfect performance validates the effectiveness of our dual-model architecture in achieving instantaneous, system-level interruption.

Table 2: Success rates on interactive tasks (30 trials).

| Interactive Task | Speech Interruption | Task Switching | Emergency Stop |
|---|---|---|---|
| **Success Rate (SR)** | 100% | 93.3% | 100% |

In the more complex task switching scenario, VITA-E achieves a 93.3% success rate. The few fail-ures are not due to the interruption mechanism itself, but stem from occasional misinterpretations by

the VLM. In these cases, the VLM fails to recognize the new user directive as an action command, consequently providing only a verbal response instead of switching its physical task. We believe this problem can be addressed by further expanding the VLM's training with more diverse embodied scenarios. These results nonetheless strongly demonstrate the superiority of our framework in handling dynamic, real-time user interventions.

## 4.3 ABLATION STUDIES

To validate the effectiveness of our fine-tuning strategy for teaching the VLM to act as a system controller, we conduct an ablation study. We compare our fine-tuned **VITA-E VLM** against its base model, **VITA-1.5** (Fu et al., 2025), on its ability to generate correct responses to user instructions like a robot. The models are prompted to act as a robot, and the accuracy of their responses, as judged by human evaluators, is shown in Table 3.

Table 3: Ablation study: Accuracy of the VLM in generating appropriate responses and control tokens before and after fine-tuning.

| Model | Cannot Execute | Exec. Inst. 1 | Exec. Inst. 2 | Emergency Stop | Task Completed |
|---|---|---|---|---|---|
| VITA-1.5 (Base) | 75% | 10% | 5% | 0% | 15% |
| **VITA-E VLM (Ours)** | **90%** | **95%** | **95%** | **100%** | **60%** |

The results show a sharp contrast between the base model and our fine-tuned version. The base VITA-1.5 model can correctly identify nonexecutable instructions with 75% accuracy, but it performs poorly in generating valid action commands and completely lacks the ability to stop the action. Common failure modes include: 1) explicitly refusing to perform actions with responses like "I cannot interact with the physical world," 2) failing to adopt the specified robot persona, and 3) only describing the steps of a plan instead of interacting as a robot.

After fine-tuning on our synthetic dataset, the VITA-E VLM's ability to interpret embodied instructions improves dramatically. It learns to refuse impossible commands with higher accuracy (90%) and, more importantly, to generate correct action instructions for executable tasks, with accuracy increasing from under 10% to 95%. Crucially, the model learns the concept of interruption from scratch, with its accuracy of emergency stop increasing from 0% to 100%. This study demonstrates that our targeted fine-tuning is essential for bridging the gap between a general-purpose VLM and a specialized "model-as-controller" that can reliably control the behaviors of our interactive system.

## 5 CONCLUSION AND FUTURE WORK

Current embodied systems face significant challenges in human-robot interaction, including rigid working patterns and the inability to handle interruptions. The VITA-E framework addresses these limitations through its innovative dual-model architecture and special token-based control flow, taking an important step toward achieving concurrent and interruptible human-robot interaction. This is enabled by our proposed "model-as-controller" paradigm, where the VLM generates its own control tokens to directly control system behavior, tightly coupling high-level reasoning with system execution. While this dual-model architecture provides robustness and an effective solution to interruption, we acknowledge that it comes at the cost of higher computational resource consumption compared to single-model systems. There are also several limitations in our current implementation: the further enhancement of model capabilities or verification of framework universality.

Although more work remains, this work opens several exciting avenues for future research. First, our architecture could be extended to handle long-horizon, multi-stage tasks by exploiting the high-level VLM to direct the low-level execution policies step-by-step. Additionally, the framework's inherent support for interruption is well-suited for incorporating real-time human feedback to correct erroneous actions or guide novel behaviors, thereby improving task success rates. Finally, while our current task-switching relies on a safe retraction to a neutral state, we plan to explore methods for achieving smoother and more efficient transitions between tasks.

We envision that this work will inspire future research toward more natural and intelligent human-robot collaboration, ultimately realizing the goal of seamless embodied assistants that can adapt and respond to human needs in dynamic, real-time environments.

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

# 6 APPENDIX

## 6.1 USE OF LARGE LANGUAGE MODELS

We thank large language models for polishing the manuscript's language, suggesting editorial revisions, and assisting with coding. All outputs are reviewed, verified, and integrated by the authors, who take full responsibility for the content and any remaining errors.

## 6.2 IMPLEMENTATION DETAILS

**Training Hyperparameters** We train our VITA-E model in two stages. First, we finetune the VITA-1.5 model on the embodied scenario vision-language data using the DeepSpeed with ZeRO-3 configuration. Then, we follow GR00T to train our model on the Libero simulation environment or the collected real-robot data. In terms of the Libero simulation environment, we first pretrain it on Libero-90, and then finetune it on the mixture of four task suites of Libero-10. The hyperparameters we used to train our VITA-E model are listed in Table 4.

Table 4: VITA-E training hyperparameters.

| Hyperparameters | Value |
|---|---|
| batch size | 64 |
| gradient accumulation steps | 1 |
| learning rate | 1e-4 |
| optimizer | AdamW |
| learning rate schedule | cosine decay |
| warmup ratio | 0.05 |
| training steps | 20000 |

**Model Hyperparameters** We listed the key parameters in our VITA-E model design in Table 5. Most model hyperparameters follow those in GR00T to ensure a fair comparison.

Table 5: VITA-E model hyperparameters.

| Hyperparameters | Libero | Real Robot |
|---|---|---|
| top image | 224×224 | 224×224 |
| wrist image | 224×224 | - |
| input state dim | 7 | 26 |
| output action dim | 7 | 26 |
| history length | 1 | 1 |
| future action prediction | 16 | 16 |
| tune visual | False | False |
| tune LLM | False | False |
| tune diffusion | True | False |
| tune projector | True | True |

## 6.3 PROMPTS

In this section, we present the prompt used to generate synthetic vision-language data for fine-tuning the VLM model as detailed in Table 6 to 9. We synthesize four categories of instruction-answer pairs to simulate the robot's responses to various user instructions, including: performing an action, being unable to complete an instruction, emergency stop, and task completed. After generating the data, we manually insert special tokens at the required locations.

<Image>

Please act as the robotic arm shown in the image. There are several objects on the table in front of you. Your task is to generate 3 different operation instructions based on these objects, and provide a corresponding robot response for each instruction.

Before giving the instructions, analyze the attributes, positions, colors, and shapes of the manipulable objects to describe and locate them more precisely. However, do not output the analysis process.

Instructions should sound natural and appropriate, and the operations must comply with the physical properties and spatial relationships of the objects.

If the instruction is unambiguous, keep it as concise as possible by omitting unnecessary details such as color, material, or relative position, for example: "Pick up the bottle", "Open the drawer", or "Place the plate on the left side of the cabinet". Avoid using vague descriptions such as "in the middle/center of the table", "near", "beside", or "next to", as these could apply to many objects. Instead, use precise relative positioning, such as "to the left front of an object", "on top of an object", "between object A and object B", "to the right back of an object", or "behind an object".

If the instruction is ambiguous, describe the object as precisely as possible, for example: "Pick up the black bottle to the right of the plate", or "Take the apple from the plate and place it on the cabinet to the right". Similarly, avoid vague descriptions like "pick up the thing on the table".

After each instruction, provide a more specific robot response. The response can be as varied and personal as possible. The response could start with a human-like phrase such as "I will pick up", "I will take", "I will help you", or "I will close", and then clearly state the object name, possibly including additional spatial details to help locate it.

Your task:

Generate 3 different operation instructions and corresponding robot responses. Instructions can involve a single object, such as "Pick up the cola", or a combination of multiple objects, such as "Pick up the apple from the table and put it on the plate". Please ensure that the objects involved actually exist in the image and that the operations are physically feasible.

For each task, please follow the format below, and output the content of the Instruction and the Response in Chinese:

Start Task <task id>

Instruction: ...

Response: ...

End Task <task id>

Table 6: Prompt for constructing action instructions and robot responses data.

<Image>

Please act as the robotic arm shown in the image. There are several objects placed on the table in front of you. Your task is to generate 3 different operation instructions that the robot will refuse to execute based on these objects, and provide a corresponding robot response for each instruction.

Before giving the instructions, analyze the attributes, positions, colors, and shapes of the manipulable objects to describe and locate them more precisely. However, do not output the analysis process.

Each instruction should either involve an object not present in the image or describe an action that is physically impossible, so the robot cannot execute it.

If the instruction is unambiguous, keep it as concise as possible by omitting unnecessary details such as color, material, or relative position, for example: "Pick up the bottle", "Open the drawer", or "Place the plate on the left side of the cabinet." Avoid using vague descriptions such as "in the middle/center of the table", "near", "beside", or "next to", as these could apply to many objects. Instead, use precise relative positioning, such as "to the left front of an object", "on top of an object", "between object A and object B", "to the right back of an object", or "behind an object".

If the instruction is ambiguous, describe the object as precisely as possible, for example: "Pick up the black bottle to the right of the plate", or "Take the apple from the plate and place it on the cabinet to the right". Similarly, avoid vague descriptions like "pick up the thing on the table".

After each instruction, provide a more specific robot response. The response can be as varied and personal as possible. The response can start with a human-like phrase, such as "I don't see...", "Sorry, ... does not exist", "I don't see...", "There is no ... on the table", "I can't...", "... cannot be...", etc., clearly stating the object name, and could include some spatial details.

Your task:

Generate 3 different operation instructions that the robot will refuse to execute and the corresponding robot responses. Instructions can involve a single object, such as "Pick up the cola", or a combination of multiple objects, such as "Pick up the apple from the table and put it on the plate". Please ensure that at least one of the objects involved in the instruction does not exist in the image, or the operation is physically impossible.

For each task, please follow the format below, and output the content of the Instruction and the Response in Chinese:

Start Task <task id>

Instruction: ...

Response: ...

End Task <task id>

Table 7: Prompt for constructing unfulfillable action instructions and robot responses data.

<Image>

Please act as the robotic arm shown in the image. You are currently performing an operational task. Generate new instructions to interrupt the ongoing task. The instructions should be as diverse and concise as possible, such as "Stop", "Terminate", etc.

After each instruction, provide a more specific robotic response. The responses should also be as varied and personalized as possible, such as "Understood, I will end the current operation", "I will immediately pause the task", or "Received, aborting the current process", etc.

Your task:

Generate 3 different instructions to interrupt the robot's operation, along with corresponding robotic responses. The instructions and responses should be natural and suitable for a real robot-human interaction.

For each task, please follow the format below, and output the content of the Instruction and the Response in Chinese:

Start Task <task id>

Instruction: ...

Response: ...

End Task <task id>

Table 8: Prompt for constructing emergency stop instructions and robot responses data.

<Image>

Please act as the robotic arm shown in the image. You have already completed an operation instruction. The image shows the scene after the operation instruction is completed. Please infer what instruction you have completed.

Before giving the instructions, analyze the attributes, positions, colors, and shapes of the manipulable objects to describe and locate them more precisely. However, do not output the analysis process.

Instructions should sound natural and appropriate, and the operations must comply with the physical properties and spatial relationships of the objects.

If the instruction is unambiguous, keep it as concise as possible by omitting unnecessary details such as color, material, or relative position, for example: "Pick up the bottle", "Open the drawer", or "Place the plate on the left side of the cabinet". Avoid using vague descriptions such as "in the middle/center of the table", "near", "beside", or "next to", as these could apply to many objects. Instead, use precise relative positioning, such as "to the left front of an object", "on top of an object", "between object A and object B", "to the right back of an object", or "behind an object".

If the instruction is ambiguous, describe the object as precisely as possible, for example: "Pick up the black bottle to the right of the plate", or "Take the apple from the plate and place it on the cabinet to the right". Similarly, avoid vague descriptions like "pick up the thing on the table".

Your task:

Generate one instruction that has already been completed. Instructions can involve a single object, such as "Pick up the cola", or a combination of multiple objects, such as "Pick up the apple from the table and put it on the plate". Ensure that the objects involved truly exist in the image and that the operation has been completed.

For each task, please follow the format below, and output the content of the Instruction in Chinese:

Start Task <task id>

Instruction: ...

End Task <task id>

Table 9: Prompt for constructing action instructions data that has been completed.