# OpenReview forum: "VITA-E: A Dual-Model Framework for Real-Time, Interruptible, and Concurrent Human-Robot Interaction"
_ICLR.cc/2026/Conference — Submitted to ICLR 2026_

### Official Review · Reviewer_Y9Vp · 2025-11-01

**Soundness:** 3
**Presentation:** 3
**Contribution:** 2
**Rating:** 6
**Confidence:** 3

**Summary:**

This paper explores interruptible and concurrent human–robot interaction using a dual-model framework, called VITA-E. Specifically, two VLMs operate as an “active model” and a “listening model”, allowing one to intervene in the other, thereby enabling interruption while the robot is executing controls. The VLM can also produce speech commands and emit special control tokens to realize a model-as-controller paradigm. The experiments show that the framework enables interruption during robot control.

**Strengths:**

1) The work studies a significant and practical challenge in HRI and generalist robotics: enabling interruptions and concurrent speech–control behavior during manipulation tasks.

2) The dual-VLM system is reasonable, and the overall interaction modes and response modes are clear and useful.

3) Real-time interruption and concurrency are crucial for HRI. The real-robot experiments demonstrate effectiveness for interruption, concurrency, and emergency stop.

**Weaknesses:**

1) The paper claims related works lack interruption during robot action, leading to limited baselines for interactive tasks. It would be useful to provide simple baselines—for example, a single-VLM system that runs continuously: If the human issues a new command when the action expert is controlling, VLM immediately sends the new command to the action expert (i.e., one fast “active VLM” that always responds with the latest command).

2) The method appears highly dependent on the VITA model, and the methodology for enabling interruption seems inspired by VITA, which may reduce the novelty. Please highlight the key differences beyond fine-tuning action experts.

3) Because the approach runs two VLMs, it is important to report memory and time usage and compare to baselines. If the dual setup reduces speed, it may hinder applicability.

4) Figure 4 shows the proposed method underperforms the GR00T baseline, especially on LIBERO-10. It is important to analyze why. This is a weakness.

**Questions:**

1) Please provide baselines for interactive tasks (e.g., a single-VLM polling/time-sliced baseline).
2) Clarify the contribution over VITA—what is fundamentally new here?
3) Report time and memory costs, and compare to baselines.
4) Why does the method underperform GR00T on LIBERO-10? Any ways to improve it (e.g., training data, partial fine-tuning, or control-token robustness)?

Please see the [Weakness] section for more details.

---

> ### Author Response · Authors · 2025-11-24
>
> We sincerely thanks for your positive recognition of our work. Our point-by-point responses to all your comments are provided below.
>
> ---
>
> **W1 & Q1: Please provide baselines for interactive tasks (e.g., a single-VLM polling/time-sliced baseline).**
>
> **R1:** Thank you very much for your suggestion. We have added two sets of experiments to quantitatively analyze the performance of VITA-E in interactive tasks and constructed a "Single-VLM Polling" system as a baseline for comparison.
>
> **1. Verification of Base Model Response Speed**
>
> First, we verified the end-to-end response speed of the VITA-1.5 model, which serves as the foundation for VITA-E. We compared it with a mainstream cascaded pipeline of similar parameter size (Whisper Small + VITA-1.5 + CosyVoice-300M). In this pipeline, the user's speech is first transcribed into text by the Whisper model, then the transcribed text and images are input into the VITA-1.5 model to generate a text response, which is finally converted into speech by the CosyVoice model. The results of this comparison are as follows:
>
> | Module | VITA-1.5 (End-to-End) | Pipeline (Whisper-VITA-CosyVoice) |
> | :--- | :--- | :--- |
> | ASR | - | 0.061s |
> | VLM Inference | 0.740s | 0.330s |
> | TTS Generation | 1.444s | 3.172s |
> | **Total Time** | **2.184s** | **3.563s** |
>
> Since the input for VITA-1.5 in the pipeline is text generated by ASR, while the input for VITA-1.5 in VITA-E is speech, there is a difference in the number of input and output tokens, as shown in the table below:
>
> | Model | Input | Image Tokens | System Prompt Tokens | Average Audio/Text Tokens | Average Output Tokens |
> | :--- | :--- | :--- | :--- | :--- | :--- |
> | VITA-E | Audio+Image | 256 | 151 | 56 | 61.6 |
> | Whisper-VITA-CosyVoice | Text+Image | 256 | 151 | 13.0 | 20.0 |
>
> The results indicate that the inference speed of the end-to-end VITA-1.5 model used in VITA-E is comparable to that of Whisper + VITA-1.5 (when converted to the same number of tokens), but the TTS generation speed is significantly faster than CosyVoice, which uses a FlowMatching architecture. Therefore, our VITA-1.5 model is faster in response speed than the Whisper+LLM+CosyVoice Baseline with approximate parameter sizes. Consequently, in the subsequent baseline comparison, the "Single-VLM" baseline we constructed is also based on the efficient architecture of VITA-1.5 to ensure fairness and exclude interference caused by performance differences in the models themselves.
>
> **2. Interactive Performance Comparison: VITA-E vs. "Single-VLM Polling + Time Slicing" Baseline**
>
> We conducted tests on a unified A6000 GPU platform. The key timing parameters were measured as follows: VLM text generation speed is 43 tokens/s; single VLA action inference takes 0.08s; time from speech end to VAD end is 0.7s; time from VAD end to start of LLM generation is 0.2s; average LLM generation time is 1.1s (approx. 100 tokens); TTS first-packet latency is 0.2s.
>
> We compared the response time differences between the "Single-VLM Polling + Time Slicing" method (Baseline) and our VITA-E method in handling interruption scenarios across two interruption scenarios and one concurrency scenario. The Baseline system uses a single VLM to process speech requests and action generation in turn, attempting to switch immediately when a new instruction is received.
>
> | Scenario | Interactive Metric | Baseline (Single-VLM) | VITA-E (Ours) |
> | :--- | :--- | :--- | :--- |
> | **Speech Interruption** | Response Latency | 2.2s | 2.2s |
> | **Action Switching** | Action State | **Severe Action Stuck (~1.1s)** | **Smooth (No Action Stuck)** |
> | **Action Switching** | Response Latency | 2.1s | 2.0s |
> | **Concurrency** | Action State | **Severe Action Stuck (~1.1s)** | **Smooth (No Action Stuck)** |
> | **Concurrency** | Voice Response Latency | 2.1s | 2.0s |
>
> Please see the next comment for a detailed analysis.

---

> ### Author Response · Authors · 2025-11-24
>
> * **Scenario A: Interruption during Voice Reply (Pure Voice Interaction).** In this scenario, since there are no physical actions competing for inference resources and speech generation is completed asynchronously by the TTS module without occupying the main VLM thread, the performance of VITA-E and the Baseline is essentially the same (approx. 2.2s). This demonstrates that a single model is sufficient for non-embodied tasks.
> * **Scenario B: Interruption during Action Execution with Task Switching.** When it is necessary to "retract the old action and execute a new action," the Baseline must wait for the current action inference step (0.08s) to complete before releasing resources to the VLM for parsing the new instruction, making it approximately 0.1s slower than VITA-E. More importantly, during the 1.1s it takes for the VLM to generate the new response, the Baseline cannot continue to output control signals for the old action. This causes the old action to terminate prematurely and become "stuck" in mid-air until the new instruction parsing is complete. In contrast, VITA-E can transition smoothly to the retraction state.
> * **Scenario C: Concurrent Voice Reply during Action Execution.** This scenario best highlights the advantages of VITA-E. In the Baseline, because a single model cannot perform "action inference" and "text generation" simultaneously, when the user asks a question, the Baseline must pause action generation to answer. This results in a complete loss of control signals for the robot during the 1.1s of LLM text generation, leading to a significant "Action Stuck" phenomenon, which is unnatural and potentially dangerous in embodied operations. In VITA-E, thanks to the dual-model architecture, the Active Model continues to output action control signals at high frequency, while the Listening Model processes the voice response in parallel. The results show that VITA-E achieves a fast voice response of 2.0s while maintaining absolute smoothness of action, achieving true "speaking while acting."
>
> In summary, although the difference in pure response speed values appears small, VITA-E solves the "Action Stuck" problem that the Baseline cannot overcome in the critical metrics of embodied intelligence: action continuity and concurrency.

---

> ### Author Response · Authors · 2025-11-24
>
> **W2 & Q2: Clarify the contribution over VITA—what is fundamentally new here?**
>
> **R2:** We appreciate the reviewer raising this critical question. We fully understand the concern that since VITA has already achieved voice interaction, VITA-E might seem like a simple fine-tuning or application extension of VITA to robotics. However, we wish to clarify that VITA-E is not merely an extension but a novel system architecture designed specifically to address the unique spatiotemporal constraints, safety requirements, and concurrency challenges of "embodied interaction". VITA addresses the interaction of "information flow", whereas VITA-E addresses the control of "physical flow".
>
> 1.  **Differences in Model Inference Paradigms:** VITA operates on a standard autoregressive generation mode. For user speech input, VITA generates text or speech tokens through a single `generate()` process. This is a relatively static input-output process. VITA-E must handle long-horizon physical actions specific to embodied models. Robot action execution is not generated in a single pass but is a high-frequency real-time feedback loop. In VITA-E, the VLA model needs to continuously receive updated visual observations and proprioceptive states and adjust its policy in real-time across multiple `forward()` inference steps.
> 2.  **Proposal of the "Model-as-Controller" Paradigm:** The output of VITA is limited to text and speech content. In VITA-E, to enable the VLM to possess system control capabilities, we designed a set of new system-level special tokens. These tokens are not just text labels; they serve directly as state machine triggers for the system. In VITA-E, the large model is not only a generator but also the highest-priority system controller. For instance, the model must learn to predict `[END]` to actively terminate an action, a capability not possessed by traditional VLA models or the VITA model.
> 3.  **Support for Heterogeneous Modal Concurrency (See, Listen, Speak, and Act simultaneously):** VITA achieves full-duplex voice, handling the interruption of speech streams. It deals with conflicts within a single modality (speech vs. speech). VITA-E addresses the concurrency of physical action and voice interaction. When the robot performs complex physical tasks, VITA-E allows the user to ask questions simultaneously. At this time, the Active Model continues to focus on high-frequency action control, while the Listening Model processes the voice response in parallel. This requires the system to design complex semaphores and mutex mechanisms at the lower level to manage model states and inter-communication, a system engineering challenge that a single generation task model like VITA does not need to consider.
> 4.  **Support for Action Retraction and Emergency Stops Designed for Embodied Safety:** "Interruption" in VITA merely means stopping TTS playback or text generation, which has no physical consequences. "Interruption" of robot actions involves safety in the physical world. If generation simply stops, the robotic arm might hang in a dangerous position. In VITA-E, we designed a specific action retraction mechanism. When a task switch occurs, the system must not only stop the current action generation but also utilize a stack to record and execute reverse motion operations to safely reset the robot. This maintenance and rollback of physical world states is a unique contribution of VITA-E.
>
> In summary, VITA-E solves the new challenges of real-time control, multi-task concurrency, system-level scheduling, and physical safety encountered when introducing these capabilities into the physical world. We believe this represents a significant leap from "multi-modal dialogue models" to "real-time interactive embodied agents."

---

> ### Author Response · Authors · 2025-11-24
>
> **W3 & Q3: Report time and memory costs, and compare to baselines.**
>
> **R3:** Thank you for your question. Regarding time costs, we provided a detailed explanation in our response to the first question. We explain the computational and memory costs as follows. The dual-model architecture indeed increases computational costs to a certain extent to achieve the required smooth interaction, but simultaneous inference for both models is only required in a minority of situations. The possible scenarios are as follows:
>
> 1.  **Idle State:** When no user instruction is received, both models are in standby mode. At this time, only the Voice Activity Detection (VAD) module is working, occupying minimal computational resources.
> 2.  **Single Instruction:** After the user issues an instruction, the "Active" model performs inference first, then decides whether to enter the "Action" state based on whether the result is a voice-only reply or requires action execution. However, in either case, only one model is required for inference. Therefore, the computational cost is no different from that of a general VLA model.
> 3.  **Interruption/Concurrency:** Only when the "Active" model is generating a voice response or executing an action and the user issues another instruction simultaneously does the "Listening" model need to perform inference at the same time. This is the only scenario where computational resources for both models are required simultaneously.
>
> In summary, additional computational costs are introduced only during the process of handling user interruptions or concurrency. However, this is the cost required to achieve the interruption and concurrency capabilities that the baseline cannot perform. For this reason, we suggest deploying the two models on two separate GPUs to avoid computational time delays caused by parallel inference.
>
> Additionally, we acknowledge that compared to the baseline, double the VRAM usage is indeed required to support the dual models.
>
> ---
>
> **W4 & Q4: Why does the method underperform GR00T on LIBERO-10? Any ways to improve it (e.g., training data, partial fine-tuning, or control-token robustness)?**
>
> **R4:** Thank you very much for your feedback. The slightly lower performance of our method on the LIBERO simulation dataset compared to GR00T is primarily due to the training strategy. GR00T conducts extensive pre-training on the VLM part, making it highly adaptable to manipulation tasks. In contrast, the VLM base used in VITA-E is derived from VITA-1.5, and its training data does not contain robot manipulation data. Compared to other LIBERO metrics, LIBERO-10 consists of long-sequence tasks, which impose higher requirements on the model's adaptability to manipulation tasks, resulting in relatively lower evaluation results.
>
> The core purpose of this method is to validate the effectiveness of the "dual-model architecture" in achieving concurrency and interruptible interaction, rather than solely pursuing State-of-the-Art (SOTA) manipulation success rates. Although freezing the VLM limits its fine-grained understanding of specific spatial relationships (leading to weaker performance on LIBERO-10 tasks), it maximally preserves the model's general semantic understanding and conversational capabilities.
>
> We strongly agree with your suggestions and will consider methods such as partial fine-tuning, expanding training data, and joint pre-training to further improve model capabilities in future work.

---

### Official Review · Reviewer_rzx1 · 2025-11-01

**Soundness:** 2
**Presentation:** 2
**Contribution:** 2
**Rating:** 4
**Confidence:** 3

**Summary:**

This paper presents VITA-E, a dual framework designed to enable flexible, real-time human-robot interaction in VLA. The core innovation is a parallel architecture where two VLA instances operate as an "Active Model" and a "Listening Model," allowing one to instantly intervene in the other. The framework enables several interaction modes. Experiments on Fourier GR2 demonstrate 100% success rates on emergency stops and speech interruptions, with 93.3% success on task switching.

**Strengths:**

- The paper targets on an important limitation in current VLAs, unabling to handle real-time interactions such as interruptions.

**Weaknesses:**

- While the research problem is interesting, the individual modules are not particularly innovative.
- As for the listening model, what is the difference between equipping traditional VLAs with an audio encoder/translator? For example, when receiving audio inputs, perhaps use an audio transcription model to translate the audio into some textual instructions for VLAs. This could at least be a baseline.

**Questions:**

- If we only tune the listening model, will the performance in Figure 4 drop as well?
- More training details should be provided, such as computational cost comparisons.
- Assume the person says a long sentence. Will the model react during this sentence? Or just wait until it is finished.

---

> ### Author Response · Authors · 2025-11-24
>
> Sincerely thanks for your efforts in reviewing this work. We hope the detailed responses help clarify your concerns.
>
> ---
>
> **W1: While the research problem is interesting, the individual modules are not particularly innovative.**
>
> **R1:** We appreciate the reviewer's affirmation of the research problem we aim to address. Indeed, we adopted mature VLMs and Diffusion Action Experts at the component level. However, we would like to emphasize that the core contribution of this paper lies not in inventing new visual encoders or action policy networks, but in proposing a novel system-level architecture and interactive paradigm. This addresses the challenge that a simple stacking of existing high-performance modules cannot solve—namely, achieving truly fluid, real-time, and interruptible embodied human-robot interaction.
>
> **1. Architectural Innovation**
> Most existing VLA systems utilize a single-threaded sequential execution architecture, which is the fundamental reason for the current rigid robot interaction and inability to respond to interruptions. We innovatively proposed a dual-model architecture based on the bionic "two hemispheres" concept. By designing precise synchronization primitives and mutual exclusion mechanisms, we assign the two models the roles of "Active" and "Listening" respectively. This architectural innovation allows the system to leverage existing modules to achieve functionalities that were previously impossible: performing voice interaction while executing physical actions, as well as fluid and natural task switching and interruption. This is pioneering at the system design level.
>
> **2. Control Paradigm Innovation**
> Traditional VLA models typically only output action trajectories. Through special token designs (e.g., `[ACT]`, `[HALT]`, `[END]`), we utilize the VLM as a controller for the entire robot system. This design establishes a new control mode where the large model can directly drive the system's state transitions. This transcends simple algorithm fine-tuning and represents a new embodied control paradigm.
>
> **3. The First Concurrently "See, Listen, Speak, and Act" System**
> Fluid human-robot interaction is a pain point in the field of embodied intelligence. Current VLA models mainly focus on the success rate of action execution while ignoring the dynamics of interaction. To the best of our knowledge, VITA-E is the first embodied system capable of simultaneous "Visual Perception, Speech Recognition, Voice Response, and Action Execution" within a unified framework. We demonstrated that through rational architectural design, the naturalness and real-time performance of interaction can be significantly improved.

---

> ### Author Response · Authors · 2025-11-24
>
> **W2: As for the listening model, what is the difference between equipping traditional VLAs with an audio encoder/translator? For example, when receiving audio inputs, perhaps use an audio transcription model to translate the audio into some textual instructions for VLAs. This could at least be a baseline.**
>
> **R2:** We thank you for pointing out this critical issue. In fact, the role played by the Hearing Model in our framework is exactly that of "equipping traditional VLAs with an audio encoder/translator" as you mentioned. We would like to first explain the necessity of our approach and then provide a relevant baseline experiment.
>
> First, we envision the audio module having the following functions:
> 1. Translating user speech into text instructions that can be understood and executed by VLAs.
> 2. Understanding user speech instructions and generating voice responses.
> 3. Generating special tokens for control based on different user intents.
> 4. Listening to user voice input in real-time to respond to the latest user requests.
>
> We merged the audio encoder/translator that would originally need to be equipped for VLAs into a single Voice VLM possessing the above four functions, and this Voice VLM happens to be part of the VITA-E VLA model. We believe this is precisely the core innovation and ingenuity of our proposed framework.
>
> Therefore, we consider the "VITA-1.5 Base" model mentioned in the ablation study in Section 4.3 (which is a Voice VLM model possessing the above four functions) to be the audio encoder/translator you referred to. Our ablation study results (Table 3) indicate that if the VITA-1.5 Base model is used without any fine-tuning, its accuracy in judging various user instructions is extremely low, and it cannot reply to user action instructions with the tone of a robotic assistant.
>
> Additionally, we verified the end-to-end response speed of the VITA-1.5 model upon which VITA-E is based. We compared it with a mainstream cascaded pipeline of similar parameter size (Whisper Small + VITA-1.5 + CosyVoice-300M). In this pipeline, user speech is first transcribed into text by the Whisper model, then the transcribed text and image are input into the VITA-1.5 model to generate text responses, which are finally converted into speech by the CosyVoice model. The results of this comparison are as follows:
>
> | Module | VITA-1.5 (End-to-End) | Pipeline (Whisper-VITA-CosyVoice) |
> | :--- | :--- | :--- |
> | ASR | - | 0.061s |
> | VLM Inference | 0.740s | 0.330s |
> | TTS Generation | 1.444s | 3.172s |
> | **Total Time** | **2.184s** | **3.563s** |
>
> Since the input to VITA-1.5 in the pipeline is text generated by ASR, while the input to VITA-1.5 in VITA-E is audio, the number of input and output tokens differs, as shown in the table below:
>
> | Model | Input | Image Tokens | System Prompt Tokens | Average Audio/Text Tokens | Average Output Tokens |
> | :--- | :--- | :--- | :--- | :--- | :--- |
> | VITA-E | Audio+Image | 256 | 151 | 56 | 61.6 |
> | Whisper-VITA-CosyVoice | Text+Image | 256 | 151 | 13.0 | 20.0 |
>
> The results show that the inference speed of the end-to-end VITA-1.5 model used in VITA-E is similar to that of Whisper + VITA-1.5 with similar parameter sizes (when converted to the same number of tokens), but the TTS generation speed is significantly faster than CosyVoice which uses a FlowMatching architecture. Therefore, our VITA-1.5 model is faster in response speed than the Whisper + LLM + CosyVoice Baseline with approximate parameter sizes.
>
> ---
>
> **Q1: If we only tune the listening model, will the performance in Figure 4 drop as well?**
>
> **R3:** Thank you for your question. However, we believe there is some ambiguity in your question. In fact, our Listening model and Action model share the same VLM Backbone. We will first re-introduce the model's training strategy and then discuss it from two perspectives. First, our VITA-E model adopts a two-stage training strategy. The first stage only trains the VLM to enable it to output special tokens and answer questions about embodied scenarios. In the second stage, we combine the VLM with the pre-trained Diffusion Action Expert from GR00T and train the model end-to-end to predict actions. In the second stage, we freeze the VLM Backbone.
>
> 1.  If you mean only adjusting the VLM part in the first stage and then directly combining it with GR00T's Diffusion Action Model, this model will be unable to realize action prediction because the intermediate features cannot be shared.
> 2.  If you mean fixing the Diffusion Action Expert in the second stage and only fine-tuning the VLM Backbone, the VLM would likely lose its general question-answering capabilities and the ability to output special tokens.
>
> In summary, we believe this question cannot be answered effectively as we cannot adjust only the Listening model.

---

> ### Author Response · Authors · 2025-11-24
>
> **Q2: More training details should be provided, such as computational cost comparisons.**
>
> **R4:** Thank you for pointing this out. Our VITA-E model adopts a two-stage training strategy, and we provide the computational cost comparisons below:
>
> 1.  **Stage 1:** Only the VLM is trained to enable it to output special tokens and answer questions about embodied scenarios. This stage uses 237K general data + 128K embodied scenario data for training, both in image + speech/text QA format. We freeze the Vision Encoder, Vision Adapter, and Audio Encoder, and only train the LLM and Audio Adapter. Training takes ~30 hours on 8 H20 GPUs.
> 2.  **Stage 2:** The VLM Backbone is fixed, and only the Diffusion Action Expert is fine-tuned to train the model end-to-end for outputting actions. This stage uses 3K embodied action data and takes ~12 hours on 1 H20 GPU.
>
> To the best of our knowledge, our training volume in the first stage is far less than other full VLM or VLA pre-training, and the fine-tuning of the Action Expert in the second stage is basically on par with conventional fine-tuning. For example, GR00T N1 end-to-end pre-training requires approximately 1000 H100s for ~48 hours. SmolVLA pre-training + fine-tuning requires approximately 30K GPU Hours. The Action Expert fine-tuning time given in the Readme for GR00T and $\pi_0$ is around 10 GPU Hours.
>
> ---
>
> **Q3: Assume the person says a long sentence. Will the model react during this sentence? Or just wait until it is finished.**
>
> **R5:** Thank you for your question. For a long sentence, as long as there is no significant interruption in the audio, the model will not react in the middle of the sentence but will wait until the sentence ends. This function depends on the Voice Activity Detection (VAD) module, which classifies speech segments to determine if there is active speech (i.e., performs a binary classification). It then identifies the start and end of speech through a series of rules. By setting a threshold for the duration of silence, we can adjust how long a pause is considered the end of a sentence. This threshold also affects the model's response speed: a smaller threshold means faster voice response but also a higher risk of identifying brief pauses in long sentences as the end of the sentence. In the current VITA-E, this threshold is set to 0.5s.

---

### Official Review · Reviewer_R1Ro · 2025-11-11

**Soundness:** 4
**Presentation:** 4
**Contribution:** 3
**Rating:** 8
**Confidence:** 4

**Summary:**

This paper introduces VITA-E, which is a framework for VLA systems that aims to improve interaction between the human and robot. VITA-E addresses a very relevant problem in current VLA systems, where the user is unable to interrupt or speak to the robot while it is executing an action. This is an actively studied problem, with the paper citing similar work like [RACER](https://arxiv.org/abs/2409.14674) and [Switch-VLA](https://arxiv.org/abs/2506.03574). What sets VITA-E apart is that it places heavier focus on interactivity with the robot by allowing asynchronous conversation and action.

VITA-E adapts the previous work of [VITA](https://arxiv.org/abs/2408.05211) to robotic tasks. The VITA-E architecture uses two instances of the VLA during execution, where one model acts as a "listener" and the other as an "actor". The authors train the VLA with an augmented dataset to output a set of control tokens that signal different behaviors in the system. This approach enhances interactivity with the system, as the listener can converse with the user, switch tasks, or stop execution while it is executing its current task.

The authors evaluate VITA-E on an extensive set of experiments using both simulation and real world scenarios. Results show that VITA-E improves interaction with the robot while also staying competitive in model performance on tasks.

**Strengths:**

+ **Originality**: The paper frames the problem of interactivity in a VLA system in an original and important manner. The authors apply the existing work of VITA to the different domain of human-robot interaction.

+ **Quality**: The paper is well structured; has many clear and visually appealing diagrams. Experiments and results sufficiently show value of the architecture.

+ **Clarity**: The presentation of ideas is well thought out and easy to understand.

+ **Significance**: The paper highlights how current VLA architectures lack flexible interaction with humans, and provides an innovative solution to the problem of robot interactivity during task execution.

**Weaknesses:**

+ The conclusion and future work section aptly addresses the weaknesses of the approach.
+ The dual model architecture requires ~2x more compute than a single model.
+ Interruption of tasks and task switching is not very flexible due to it currently relying on the safe retraction to neutral state.
+ There is a loss of model performance from fine-tuning, but as the authors mentioned this is because they had to freeze the base VLM.

**Questions:**

+ On page 4, you refer to the "Listening" state as "Hearing" state. I'd suggest sticking with a single term to describe this state throughout the paper.

---

> ### Author Response · Authors · 2025-11-24
>
> We sincerely thanks for your positive recognition of our work. Our point-by-point responses to all your comments are provided below.
>
> ---
>
> **W2: The dual model architecture requires ~2x more compute than a single model.**
>
> **R1:** Thank you for your comment. We would like to provide the following clarification regarding computational and memory costs. While the dual-model architecture indeed increases computational costs to some extent to achieve the required fluid interaction, it does not require up to two times the computation because simultaneous inference for both models is needed only in rare cases. The specific scenarios are as follows:
>
> 1.  When no user instruction is received, both models are in a idle state. In this case, only the Voice Activity Detection (VAD) module is working, consuming minimal computational resources.
> 2.  After the user issues an instruction, the **Active Model** first performs inference. It then decides whether to enter the **Action** state based on whether the result is a voice-only reply or requires action execution. In either case, only one model performs inference, so the computational cost is no different from a general VLA model.
> 3.  Simultaneous inference for the **Listening Model** is required only when the user issues another instruction while the **Active Model** is already generating a voice reply or executing an action. This is the only scenario where inference computation is performed for both models simultaneously.
>
> In summary, additional computational costs are introduced only when handling user interruptions or concurrency. This is the trade-off required to achieve the interruption and concurrency capabilities that baseline models cannot perform. Additionally, we acknowledge that compared to the baseline, this approach does require double the VRAM usage to support the dual models. Therefore, we recommend deploying our VITA-E system across two graphics cards to avoid increased inference latency caused by parallel inference.
>
> ---
>
> **W3: Interruption of tasks and task switching is not very flexible due to it currently relying on the safe retraction to neutral state.**
>
> **R2:** We fully agree with your observation. Currently, our VITA-E system must retract to a neutral state when handling task interruptions and task switching, which reduces the flexibility of the system. However, this design is based on the following two considerations:
>
> 1.  **Safety:** The model needs to ensure the safety of the motion path during task switching. Switching directly from an old action to a new action is highly likely to result in the generation of unpredictable action paths, which could be dangerous.
> 2.  **Generalization:** When switching from an old action to a new one, the model enters a new path that it has not been trained on. Our current training data is not yet rich enough to support the correct generalization of the model when switching between old and new inference paths. To ensure better performance, we currently adopt the design of retracting to a neutral state. This maintains consistency with the states in the training data and improves the success rate of actions.
>
> In the future, we will consider designing new algorithms and collecting action-switching data to support more flexible task switching.
>
> ---
>
> **W4: There is a loss of model performance from fine-tuning, but as the authors mentioned this is because they had to freeze the base VLM.**
>
> **R3:** Thank you very much for your comment. The issue regarding the method performing slightly lower than SOTA VLA models on the simulation dataset is primarily due to our training strategy. Other VLA models conduct extensive pre-training on the VLM part to make it highly adaptable to manipulation tasks. In contrast, the VLM base used in VITA-E is derived from VITA-1.5, and its training data does not include robot manipulation data. Compared to other LIBERO metrics, LIBERO-10 consists of long-horizon tasks, which require higher adaptability from the model for manipulation tasks, leading to relatively lower evaluation results.
>
> The core objective of this method is to validate the effectiveness of the **Dual-Model Architecture** in achieving concurrent and interruptible interaction, rather than purely pursuing SOTA manipulation success rates. Although freezing the VLM limits its fine-grained understanding of specific spatial relationships (resulting in weaker performance on LIBERO-10 tasks), it maximally preserves the model's general semantic understanding and conversational capabilities.
>
> We strongly agree with your suggestion and will consider methods such as partial fine-tuning, expanding training data, and joint pre-training to further enhance model capabilities in the future.

---

> ### Author Response · Authors · 2025-11-24
>
> **Q1: On page 4, you refer to the "Listening" state as "Hearing" state. I'd suggest sticking with a single term to describe this state throughout the paper.**
>
> **R4:** Thank you very much for pointing out this terminology issue, which can easily cause confusion. First, we need to clarify that "Listening" and "Hearing" do not refer to the same state in the paper. The VITA-E system has two corresponding sets of concepts that indicate different states of the system and the models respectively: **Active-Listening** and **Hearing-Acting**.
>
> * **Active-Listening** refers to the two identical models deployed simultaneously, where one is in an active state and the other is in a listening state. User instructions are processed by the model in the active state, while the listening model is responsible for handling interruptions and concurrency.
>
> * **Hearing-Acting** refers to the two states of the **same** active model when processing user requests. The active model defaults to the **Hearing** state, where it acts as a VLM to perform a complete forward inference (generate) on the user instruction to produce a reply and identify intents by generating special control tokens. Subsequently, if the identification result indicates that the user instruction requires completing an action, the active model enters the **Action** state. The model then acts as a VLA, continuously performing inference (forward) on the current robot state to generate and execute actions until the action is completed.
>
> We acknowledge that the terminology here is indeed highly prone to confusion. We will modify the paper to change **Active-Listening** to **Active-Standby** to avoid this confusion.

---

### Official Review · Reviewer_TVi9 · 2025-11-11

**Soundness:** 3
**Presentation:** 2
**Contribution:** 3
**Rating:** 4
**Confidence:** 5

**Summary:**

VITA-E proposes a real-time VLA framework that makes robot interaction interruptible, concurrent, and safe. It runs two VLA instances in parallel—an Active model that executes and a Listening model that can preempt—while the VLM serves as a controller by emitting special tokens (e.g., [ACT], [RES], [INST], [HALT], [END]) that drive a simple state machine for speaking vs. acting. Training reformats embodied data so the model learns to produce these control tokens for manipulation and safety. On a physical humanoid platform, VITA-E achieves fluent talk-while-act (~2.26 s voice latency), 100% success for speech interruption and emergency stop, and 93.3% success for task switching—demonstrating a practical, general recipe for fluid, interruptible human-robot collaboration.

**Strengths:**

The paper is genuinely original in how it reframes embodied interaction around preemption and safety, combining a dual-model architecture with a lightweight control-token interface to remove the “turn-based” limitation common in VLAs. Its quality is supported by thoughtfully chosen real-robot evaluations that directly test concurrency, interruption, task switching, and emergency stop, plus ablations that show the control-aware fine-tuning—not just prompting—is what yields reliability. The writing is clear and reproducible: the state machine, token semantics, and data formatting are specified precisely, with figures that make the listen/act transitions easy to follow. In significance, the work matters because it elevates interruptibility and safe handoffs to first-class capabilities, offering a model-agnostic interaction layer that others can adopt across stacks; this has immediate relevance for deployable HRI in homes, assistive settings, and light industry, even if downstream perception or control modules evolve.

**Weaknesses:**

1. Model design: The dual-model architecture introduces a trade-off between responsiveness and computational cost. The authors acknowledge this limitation, and the framework’s generalizability to broader robot stacks has not been demonstrated.

2. Evaluation scope: The experimental evaluation is narrow, especially regarding interaction performance. Concurrency results are reported qualitatively, while other metrics are based on 30-trial averages. Baselines are largely omitted due to mismatched capabilities, leaving comparative advantages insufficiently analyzed.

3. Performance: On manipulation benchmarks, the proposed method is not state-of-the-art, which may limit its appeal to researchers focused on maximizing task success.

4. Task switching: Reported failures arise from the VLM misclassifying new directives as dialogue, revealing brittleness in the control-token decision boundary under distribution shifts.

5. Training and generalization: The pipeline relies on synthetic token injection (e.g., simulated “Stop!” events) instead of real interruption data, raising questions about generalization and safety in real-world scenarios.

6. Unaddressed challenges: Core capabilities such as handling long-horizon, multi-stage tasks and ensuring smooth transitions remain future work. Consequently, while the interaction layer is promising, it is not yet a turnkey solution for complex deployments.

**Questions:**

1. What is the approximate time it takes from giving a command to stopping the action mentioned in the paper? The time is the same for different tasks？

2. Whether the success rate of emergency stop and voice interruption can be maintained on different real machines, and whether the efficiency can be maintained？

---

> ### Author Response · Authors · 2025-11-24
>
> Sincerely thanks for your efforts in reviewing this work. We hope the detailed responses help clarify your concerns.
>
> ---
>
> **W1: Model design: The dual-model architecture introduces a trade-off between responsiveness and computational cost. The authors acknowledge this limitation, and the framework’s generalizability to broader robot stacks has not been demonstrated.**
>
> **R1:** Thank you for this comment. The dual-model architecture indeed introduces a certain level of computational cost to achieve fluid interaction. However, simultaneous inference for both models is required only in specific scenarios. The operational states are as follows:
> 1.  **Idle State:** When no user command is received, both models remain on standby. Only the Voice Activity Detection (VAD) module is active, consuming minimal computational resources.
> 2.  **Single-Model Inference:** When a user issues a command, the "Active Model" first performs inference. Based on the result, it decides whether to generate a voice reply or transition to the "Action" state. In either case, only one model performs inference, making the computational cost comparable to a standard VLA model.
> 3.  **Dual-Model Inference:** Simultaneous inference is only required when the user issues a new command **while** the "Active Model" is already generating a voice response or executing an action (i.e., the "Listening Model" processes the interruption). In this scenario, traditional VLA models either cannot handle it or will cause significant latency and stuttering.
> In summary, additional computational costs are introduced only during user interruptions. We acknowledge that supporting two models requires double the VRAM usage. Therefore, we recommend deploying the VITA-E system across **two GPUs** to avoid increased latency caused by parallel inference contention.
>
> Regarding generalization, our framework can be deployed on a wide range of robotic hardware systems, provided they are equipped with speakers, microphones, and the necessary components for embodied tasks. Furthermore, our framework is model-agnostic and compatible with mainstream VLA models utilizing a "System 1 – System 2" architecture, such as $\pi_0$ and SmolVLA. The only required modification is fine-tuning the VLM component of these models to generate special control tokens. Subsequently, the entire VLA model would need end-to-end fine-tuning to realign the VLM weights for action generation. Once capable of generating special control tokens, these models can be integrated into our framework. We acknowledge that, due to hardware limitations and the complexity of retraining new models, we have not yet applied the system to other robotic hardware or migrated it to other VLA models. This work serves as a verification of the proposed system workflow, leaving broader generalization and functional expansion to future work.

---

> ### Author Response · Authors · 2025-11-24
>
> **W2: Evaluation scope: The experimental evaluation is narrow, especially regarding interaction performance. Concurrency results are reported qualitatively, while other metrics are based on 30-trial averages. Baselines are largely omitted due to mismatched capabilities, leaving comparative advantages insufficiently analyzed.**
>
> **R2:** We appreciate the reviewer pointing this out. We acknowledge the difficulty in evaluating interaction performance. First, existing VLA models are architecturally incapable of capabilities such as interruption and concurrency, making direct comparison impossible. Therefore, in Section 4.3, we conducted ablation studies comparing the model before and after fine-tuning on embodied scenarios, demonstrating that the fine-tuned model possesses superior instruction-following capabilities. We also reported an average response latency of 2.26s to demonstrate real-time performance. Second, considering the high cost of real-robot experiments, 30 trials is a standard practice in the robotics field. Combined with the Libero simulation experiments, we believe the evaluation is sufficient.
>
> To further demonstrate the advantages of VITA-E in interactive performance, we provide two additional comparative experiments:
>
> **1. Voice Response Speed Comparison:**
> We compared VITA-E against a baseline pipeline with similar parameter counts: Whisper + VITA-1.5 + CosyVoice. Specifically, we compared it with a mainstream cascaded pipeline (Whisper Small + VITA-1.5 + CosyVoice-300M). In this pipeline, user speech is first transcribed into text by the Whisper model; then, the transcribed text and images are input into the VITA-1.5 model to generate response text; finally, the response text is converted into speech by the CosyVoice model. The results of this comparative experiment are as follows:
>
> | Module | VITA-1.5 (End-to-End) | Pipeline (Whisper-VITA-CosyVoice) |
> | :--- | :--- | :--- |
> | **ASR** | - | 0.061s |
> | **VLM Inference** | 0.740s | 0.330s |
> | **TTS Generation** | 1.444s | 3.172s |
> | **Total Latency** | **2.184s** | **3.563s** |
>
> Since the input to VITA-1.5 in the pipeline is text generated by ASR, whereas the input to VITA-1.5 in VITA-E is speech, there are discrepancies in the number of input and output tokens. The specific details are shown in the table below:
>
> | Model | Input | Image Tokens | System Prompt Tokens | Avg. Audio/Text Tokens | Avg. Output Tokens |
> | :--- | :--- | :--- | :--- | :--- | :--- |
> | **VITA-E** | Audio+Image | 256 | 151 | 56 | 61.6 |
> | **Whisper-VITA-CosyVoice** | Text+Image | 256 | 151 | 13.0 | 20.0 |
>
> The results indicate that the inference speed of the end-to-end VITA-1.5 model used in VITA-E is comparable to that of the Whisper + VITA-1.5 combination with similar parameters (when normalized for token count). However, the TTS generation speed is significantly faster than CosyVoice, which adopts the FlowMatching architecture. Therefore, our VITA-1.5 model achieves faster response speeds than the baseline pipeline using similar parameters (Whisper + LLM + CosyVoice).
>
> ---
>
> **W3: Performance: On manipulation benchmarks, the proposed method is not state-of-the-art, which may limit its appeal to researchers focused on maximizing task success.**
>
> **R3:** We appreciate this feedback. The slightly lower performance on simulation datasets compared to SOTA VLA models is primarily due to our training strategy. Other VLA models undergo extensive pre-training on the VLM component to adapt it to manipulation tasks. In contrast, the VLM base used in VITA-E (VITA-1.5) does not include robot manipulation data in its training set. Compared to other Libero suites, LIBERO-10 consists of long-horizon tasks requiring higher adaptability, leading to relatively lower evaluation scores.
>
> The core objective of this method is to verify the effectiveness of the "Dual-Model Architecture" in achieving concurrent and interruptible interaction, rather than solely pursuing SOTA manipulation success rates. Although freezing the VLM limits its fine-grained understanding of spatial relationships (impacting Libero-10 performance), it maximally preserves the model's general semantic understanding and conversational capabilities. We agree with the reviewer's suggestion and will consider methods such as partial fine-tuning, data expansion, and joint pre-training to further enhance manipulation capabilities in future work.

---

> ### Author Response · Authors · 2025-11-24
>
> **W4: Task switching: Reported failures arise from the VLM misclassifying new directives as dialogue, revealing brittleness in the control-token decision boundary under distribution shifts.**
>
> **R4:** We thank the reviewer for identifying this critical detail. The 6.7% failure rate in task switching was indeed due to the VLM misclassifying new action directives as pure dialogue. However, we emphasize that this limitation stems from insufficient instruction fine-tuning data coverage rather than a flaw in the VITA-E dual-model architecture itself. Despite a small number of misclassifications, our fine-tuning strategy significantly strengthened the decision boundary. As shown in the ablation study (Table 3), the base model (VITA-1.5) had an accuracy of less than 10% in identifying executable instructions, whereas the fine-tuned VITA-E VLM achieved 95%. This proves the effectiveness of our "Control Language" fine-tuning. To address the "brittleness" and further improve accuracy, we plan to expand the dataset with a focus on collecting data from real-world scenarios to reduce distribution shifts. We believe that increasing data scale and diversity will significantly reduce such classification errors.
>
> ---
>
> **W5: Training and generalization: The pipeline relies on synthetic token injection (e.g., simulated “Stop!” events) instead of real interruption data, raising questions about generalization and safety in real-world scenarios.**
>
> **R5:** We appreciate the reviewer's concern regarding the construction and safety of training data. While synthetic data is often considered less reliable than real data, we wish to clarify why a training strategy based on synthetic tokens is the optimal choice for robust generalization of the "Emergency Stop" function.
>
> 1.  **Visual Decoupling:** In our system design, "Emergency Stop" is defined as a highest-priority semantic instruction-following task, rather than an obstacle avoidance task based on visual perception. Regardless of what the robot sees or its current pose, it must unconditionally execute the stop command upon receiving the user's voice instruction. By fine-tuning with randomly sampled image frames paired with "Stop" voice commands, the model learns to decouple the "Stop" behavior from the visual context. This prevents the model from learning spurious correlations (e.g., "stop only against specific backgrounds"), ensuring the function triggers reliably in diverse, unseen real-world scenarios.
> 2.  **Explicit Control:** Our safety mechanism relies not on the model's implicit understanding of complex physics, but on explicit system-level control. Through fine-tuning, the VLM learns a conditioned reflex: output the [HALT] token upon hearing a stop command. Once generated, the underlying state machine immediately terminates the action loop.
> 3.  **Experimental Evidence:** As shown in Table 2, VITA-E achieved a 100% success rate in real-robot "Emergency Stop" tests. Even in manipulation scenarios the model had never seen, the robot reliably generated the [HALT] token provided the voice command was clear. This strongly demonstrates the method's generalization capability in the real world.
>
> We agree that proactive stopping based on visual perception (e.g., autonomous obstacle detection) is an important future direction, and we plan to enhance system safety by incorporating visual perception modules in subsequent work.

---

> ### Author Response · Authors · 2025-11-24
>
> **W6: Unaddressed challenges: Core capabilities such as handling long-horizon, multi-stage tasks and ensuring smooth transitions remain future work. Consequently, while the interaction layer is promising, it is not yet a turnkey solution for complex deployments.**
>
> **R6:** We fully agree with the reviewer that handling long-horizon, multi-stage tasks and ensuring smooth transitions are core capabilities for complex robot deployment. We do not claim VITA-E is a turnkey solution for all manipulation challenges; rather, we position it as a significant step toward more natural and responsive robotic assistants.
>
> 1.  **Interaction Paradigm Focus:** The core contribution of this paper is the innovation of the "Interaction Paradigm" rather than the completion of "Manipulation Capabilities." The primary pain point of current VLA models is the "rigid, static interaction paradigm" (uninterruptible and non-concurrent). VITA-E overcomes this bottleneck via its dual-model architecture and special token control system. The 100% success rate in emergency stops and voice interruptions, along with the ability to handle concurrent action and speech, proves the architecture's effectiveness at the interaction layer.
> 2.  **Safety and Validation:** We acknowledge that the current action retraction mechanism, while ensuring safety, has room for improvement in fluidity. Retracting to a safe position was prioritized to ensure system safety while validating the task-switching logic. Even without smooth transitions, VITA-E successfully demonstrated the capability to be dynamically redirected from Action A to Action B, verifying the validity of the dual-model core at the logical level.
> 3.  **Foundation for Long-Horizon Tasks:** VITA-E serves as a foundational base for long-horizon tasks. While our experiments focused on atomic tasks, the architecture inherently supports extension to long-horizon tasks. Existing long-horizon planning methods often lack interruption mechanisms; VITA-E fills this gap, providing the necessary interactivity for future models to incorporate human-in-the-loop guidance and correction.
>
> ---
>
> **Q1: What is the approximate time it takes from giving a command to stopping the action mentioned in the paper? The time is the same for different tasks？**
>
> **R7:** Thank you for the question. The time mentioned can be divided into two parts:
> 1.  **Hearing Phase:** The time to identify user intent and generate a response. As mentioned in Section 4.2, the average latency over ten trials is 2.26 seconds. This time is consistent across most tasks.
> 2.  **Action Phase:** The time required to complete the physical action. This varies significantly by task and is influenced by the duration of the human demonstration and the data recording frequency (20Hz in our case).
> The average durations to complete the two actions designed in the paper are as follows:
>
> | **Action** | **(a) Pick up can** | **(b) Pick up and place toy** |
> | :--- | :--- | :--- |
> | **Time** | ~42s | ~32s |
>
> ---
>
> **Q2: Whether the success rate of emergency stop and voice interruption can be maintained on different real machines, and whether the efficiency can be maintained？**
>
> **R8:** These two points are critical for embodied intelligence systems.
> First, the success rate of emergency stop and voice interruption can be maintained across different hardware. This depends primarily on the VLM's ability to accurately judge the intent of the user's voice command, independent of the robotic hardware used. We collected a large and diverse volume of emergency stop voice data via LLM generation to ensure the model's judgment capability. For voice interruption, the VITA-1.5 model itself was trained with a massive amount of human instruction speech and noise audio (irrelevant conversation) to distinguish between them. Our fine-tuning process followed this method, ensuring high success rates in interruption response.
>
> Second, regarding system efficiency, this depends largely on the GPU used for inference. We deployed the proposed VITA-E system on a single NVIDIA A6000 GPU. System efficiency can be maintained by using GPUs with equivalent or higher computational power for model inference.

---

### Meta-Review · Area_Chair_YkUB · 2026-01-06

**Summary:**

The reviewers generally acknowledge the paper’s clarity, interesting/original problem formulation, and practical significance. These particularly include its dual-model architecture, the model-as-controller paradigm using control tokens, and real-robot validation of real-time interruption, concurrency, and emergency stopping.
However, several substantial concerns converge across reviews:

- Insufficient Evaluation. Three reviewers pointed out that the experimental scope is limited by the omission of critical baselines (e.g., the related work and simple VLM extensions), undermining claims of architectural necessity.
- Sub-optimal Performance. The method underperforms established VLA baselines on standard manipulation benchmarks, indicating a problematic trade-off between interactivity and task competence.
- Limited Novelty. Reviewers questioned the technical contribution, noting that the individual modules lack innovation and the methodology is not sufficiently differentiated from prior works like VITA.
- Computational Efficiency. The dual-model architecture introduces significant computational and memory overhead, yet the paper lacks a rigorous efficiency analysis compared to single-model implementations.

**Reviewer Concerns:**

The rebuttal clarifies implementation details but does not substantively resolve the major weaknesses.

- The authors have partially alleviated concerns regarding system latency by providing additional experimental comparisons on voice response speed and interactive performance.
However, these validate throughput, not interactive robustness nor task success under interruption.
- The performance gap on manipulation benchmarks remains unmitigated; attributing it to frozen VLM and lack of manipulation pretraining confirms a fundamental trade-off—without evidence that interactivity gains justify the loss in core capability.
- Critical concerns persist regarding the limited technical novelty compared to prior work and the significant. Rebuttal emphasizes system integration but fails to isolate conceptual or algorithmic novelty beyond repurposing VITA with control tokens and dual instantiation.
- The evaluation of interactive tasks remains too narrow to demonstrate sufficient robustness and generalization in complex, real-world interruption scenarios.

**Reviewer Scores:**

As summarized above, limited new evidence (e.g., only speed comparisons) is unlikely to shift the assessment. Some main concerns are acknowledged but deferred to future work. It is unlikely that the reviewers would change their scores.

---

### Decision · Program_Chairs · 2026-01-26

Reject